# Binge Eating Disorder—The Point Prevalence among Polish Women with Polycystic Ovary Syndrome and Validity of Screening Tool for This Population

**DOI:** 10.3390/ijerph20010546

**Published:** 2022-12-29

**Authors:** Katarzyna Suchta, Roman Smolarczyk, Krzysztof Czajkowski, Ewa Rudnicka, Andrzej Kokoszka

**Affiliations:** 1Department of Gynaecological Endocrinology, Medical University of Warsaw, Karowa 2 Str., 00-315 Warsaw, Poland; 2II Department of Obstetric and Gynaecology, Medical University of Warsaw, Karowa 2 Str., 00-315 Warsaw, Poland; 3II Department of Psychiatry, Medical University of Warsaw, 02-091 Warsaw, Poland

**Keywords:** polycystic ovary syndrome, binge eating disorder, point prevalence, screening tool

## Abstract

Background: Polycystic ovary syndrome (PCOS) is one of the most common endocrine disorders which is associated with an increased risk of metabolic dysregulation. The elevated prevalence of obesity has been observed in women with PCOS. Since obesity is commonly associated with eating disturbances, including the binge eating disorder (BED), and since the hormonal changes in PCOS patients could influence the food intake model, we decided to estimate the prevalence of BED in PCOS patients and to assess the sensitivity and specificity of the Questionnaire for Binge Eating Screening (QBES) in PCOS patients. Methods: A total of 122 hospitalized women with PCOS aged 16–45 (M = 26; SD = 5.22) took part in the study. Binge eating disorder (BED) was diagnosed according to the DSM-5 diagnostic criteria. QBES was used as a screening tool for BED. Results: The point prevalence of BED in PCOS women according to DMS-5 criteria was 51 (42%). At least two positive answers to four QBES items had 100% sensitivity and 91% specificity. Positive answers to even only the first two questions from QBES had 98% sensitivity and 85% specificity. Conclusions: Women with polycystic ovary syndrome are at a very high risk of binge eating behaviors. Screening for eating disorders should be a routine procedure in women with PCOS. The first two questions from QBES are a brief and relatively reliable screening tool that may be used in everyday practice with POSC patients.

## 1. Introduction

Polycystic ovary syndrome (PCOS) is the most frequent endocrine disorder in women of reproductive age, with a point prevalence of up to 15% [1]. Its diagnosis requires the presence of at least two out of three elements, i.e., biochemical and/or clinical hyperandrogenism, anovulatory menstrual cycles and/or rare ovulations, and a specific echogenicity of the ovaries in a pelvic ultrasound examination as well as an exclusion of other androgen excess or related disorders [2] The occurrence of PCOS is associated with the increased risk of infertility, thyroid dysfunction, anxiety, depression, sexual disorders, a subjectively lower quality of life, and, finally, with metabolic disturbances such as obesity [3,4,5,6]. The results of a meta-analysis indicate that the mean prevalence of obesity in women with PCOS is 49% [7]. Barrea et al. indicate that the type of chronotype in PCOS women can influence their eating patterns and promote obesity and metabolic consequences. According to the authors, the evening chronotype is associated with worse eating habits and with a most severe insulin resistance [8]. However, it is worth noticing that both obese and lean women with PCOS have a higher amount of visceral adipose tissue in comparison to controls and because of that are at higher risk of metabolic consequences. This is why eating habits and an appropriate diet play an essential role in treatment of PCOS [9]. A review of the literature indicates a higher prevalence of binge eating disorder (BED) in this group than in the general population, which varies from 12% to 39% [5,10,11]. In the literature, there are concepts that try to clarify connections between eating problems and PCOS. Steegers-Theunissen R.P.M. et al. [12] suggest that eating disorders, especially during puberty, which are based on body image dissatisfaction and emotional distress, can promote the epigenetic dysregulation of the hypothalamic–pituitary–gonadal axis which leads to changes in folliculogenesis. In this concept, PCOS is induced by recurrent episodes of overeating and/or dieting and psychological distress during puberty and adolescence. The authors [12] also indicate a possible role of the gut microbiota in this process, as its changes during inappropriate eating patterns can lead to the epigenetic modification of neurohormone regulation. The authors of this research suggest that this process can also be connected with the impact on the offspring of women with PCOS. It is a novel concept that requires further study. Nevertheless, it shows a connection between a PCOS risk and eating patterns and their interrelation [12]. Relations of binge eating (BED) and obesity were described in 1959 by Stunkard [13]. BED is related to obesity and its complications [14]. BED disorder was introduced in the Diagnostic Statistical Manual of Mental Disorders DSM–IV (APA, 1994) [15], but it is not included in the current version of the International Classification of Diseases and Health Related Problems, the 10th Revision (ICD-10) [16]. All of the above indicate the need to identify the factors related to the occurrence of BED, including psychological features [10,11]. The diagnosis of BED is important as it may be successfully treated with behavioral intervention, psychoeducational interventions, psychotherapy (mainly Cognitive-Behavioral Therapies—CBT; Dialectic Behavioral Therapy—DBT; or Interpersonal Psychotherapy—IPT), and pharmacotherapy. At present, there is no comprehensive and reliable study on the prevalence of BED among Polish women with PCOS. According to the authors’ best knowledge, this is the first study to have been conducted on Polish women with PCOS. Clarification of the usefulness of the Questionnaire for Binge Eating Screening (QBES) in PCOS patients is an additional goal of the study [17].

## 2. Materials and Methods

The study was conducted among women hospitalized at the Endocrinological Gynecology Department of the Medical University of Warsaw. The study was approved by the Ethics Committee of the Medical University of Warsaw (KB/172/2017), and the patients signed informed consent forms before the study began.

The diagnosis of PCOS was made using the Rotterdam criteria [2] which include: (1) clinical and/or biochemical hyperandrogenism; (2) anovulatory menstrual cycles and/or rare ovulations, and (3) a characteristic structure of the ovaries in an ultrasound examination. Clinical hyperandrogenism consists of hirsutism, acne, and/or alopecia. The degree of hirsutism was assessed by using the Ferriman–Gallwey scale. A score of 8 or above confirmed hirsutism. Biochemical hyperandrogenism consists of elevated levels of androgen hormones (testosterone and androstenedione). Anovulation was confirmed based on the serum progesterone level in the 22nd to 24th day of the cycle. A level equal to 3 ng/mL or below confirmed an anovulatory cycle. A pelvic ultrasound was performed using Aloka 7 alpha equipment to assess the morphology of the ovaries. Oligomenorrhea was defined as the duration of a menstrual cycle for over 35 days and secondary amenorrhea as a lack of menstrual bleeding for over 6 months.

The excluding criteria were the patient’s refusal to participate in the study and a diagnosis of the cause of hyperandrogenism which was different from PCOS, such as non-classical adrenal hyperplasia, an androgen-secreting tumor, or Cushing’s syndrome. An additional excluding criterion was the use of oral contraception, glucocorticoids, or biguanides for up to six months before participation in the study, as all of these drugs have an impact on hormonal function and the level of serum androgens.

Binge eating disorder was diagnosed according to the DSM-5 criteria [APA] [18]. These include: “A - Recurrent episodes of binge eating. An episode of binge eating is characterized by (A1) Eating, in a discrete period of time, an amount of food that is definitely larger than most people would eat in a similar period of time under similar circumstances. (A2) A sense of lack of control over eating during the episode (for example, a feeling that one cannot stop eating or control what or how much one is eating).B - The binge-eating episodes are associated with three (or more) of the following: (B1) Eating much more rapidly than normal. (B2) Eating until feeling uncomfortably full. (B3) Eating large amounts of food when not feeling physically hungry. (B4) Eating alone because of feeling embarrassed by how much one is eating. (B5) Feeling disgusted with oneself, depressed, or very guilty afterwards. C - Marked distress regarding binge eating is present. D - The binge eating occurs, on average, at least once a week for three months. E - The binge eating is not associated with the recurrent use of inappropriate compensatory behavior (for example, purging) and does not occur exclusively during the course of anorexia nervosa, bulimia nervosa, or avoidant/restrictive food intake disorder.”

To make a diagnosis, criteria A1 and A2 must be present with at least three or more from criteria B1 to B5. As far as frequency is concerned, the episodes must occur at least once a week during a period of at least three months. Episodes should not be associated with compensatory behaviors and usually are associated with feeling distress about the episodes. To make the diagnosis according to these criteria, a physician needs about 20min to interview the patient.

The Questionnaire for Binge Eating Screening (QBES) [17] consists of the four following questions: “(1) Does it happen rather often that you eat large amounts of food in a brief period of time? (2) When this is the case, do you feel that you lack control over eating–you cannot stop eating until you get an unpleasant feeling of being too full? (3) When you feel unwell, is eating a way for improving your mood? (4) Is binge eating a reason behind putting up on weigh, above the weight that is normal for you?”. According to the authors of the questionnaire, if the questioned person responds “yes” to more than one of the items, this indicates the presence of binge eating. The sensitivity and specificity of this screening tool has not been verified so far. QBES is a self-rating questionnaire that does not require the physician’s involvement. 

Statistical analysis was conducted using IBM SPSS Statistics (v.25) software. The point prevalence of BED among women with PCOS was assessed. The sensitivity and specificity of more than zero, one, two and three positive answers in the QBES questionnaire, as well as that of positive answers to the first two questions of the QBES questionnaire, were assessed. Sensitivity was assessed using the following model: the number of truly positive results divided by the number of truly positive results + the number of false negative results. Specificity was assessed using the following model: the number of truly negative results divided by the number of truly negative results + the number of false positive results. 

## 3. Results

A total of 122 out of 140 (87.1%) women with PCOS who had been invited to participate took part in the study. Their age ranged between 16 and 45 (M = 26; SD = 5.22). Fifty-one (51; 42%) of all women with PCOS met the DSM-5 criteria for BED, but its symptoms were present also among twenty-five (35%) persons without clinical BED. Forty-six (46; 65%) women without BED did not have any of its symptoms. Considering the descriptions of the occurrence of symptoms, BED symptoms may have an important impact on eating habits. The point prevalence of specific symptoms among women with PCOS with and without BED is presented in Table 1.

The sensitivity and specificity of the Questionnaire for Binge Eating Screening (QBES) among women with PCOS is presented in Table 2. Positive answers to all four questions identifies all cases of BED (100% sensitivity) and has very high specificity (91%). It is noteworthy that there is very high sensitivity (98%) and specificity (85%) for the positive response to the first and second questions in the Questionnaire for Binge Eating Screening. Asking these questions makes screening for BED in the PCOS population easy and very brief. The number of women with BED without at least one positive answer to the screening questions was zero. Fifty (50; 98%) women with BED answered positively to both of the first two questions of the Questionnaire for Binge Eating Screening (QBES), while among the individuals without BED in the group of PCOS women, only eleven (15%) responded in this manner.

## 4. Discussion

According to the best of our knowledge, this is the first study to evaluate the prevalence of binge eating disorder among PCOS patients in Poland. The relatively low rate of refusals indicates that this population is rather representative for the patients diagnosed and treated in the department where the research was conducted, and it is sufficient large for valid research. 

Furthermore, to the best of our knowledge, no other data exist regarding the prevalence of BED in the general Polish population. The lifetime prevalence of BED assessed in six European countries in the general population was 1.12% (CI = 0.95) [19]. In our analysis, we report that binge eating behaviors occur about ten times more frequently among PCOS women when compared to other authors’ results and about twenty times more frequently when compared to non-PCOS women. In our analysis, 42% of women with PCOS presented BED. Kessler R.C. et al., in their analysis assessing the lifetime occurrence of BED in populations of 14 countries (including Europe, North and South America, and New Zealand), reported the prevalence of BED among 1,9% women in the general population [20]. Aneesa Thannickal A. et al., in their review, reported the occurrence of BED among non-PCOS women of 1,61% and 5,42% in PCOS women [21]. Our results are comparable to publications by Jeanes Y.M. et al. [11] and Lee I. et al. [10] reporting that approximately 30% of women with PCOS have BED. Jeanes Y.M. et al. reported that approximately 20% of lean women with PCOS, 16% of overweight women with PCOS, and 37% of obese women with PCOS presented BED, while Lee et al. reported that approximately 20% of women with PCOS presented BED. It has been suggested that PCOS and binge eating share common risk factors and etiology, including common metabolic, hormonal, and mental conditions which may predesignate both PCOS and BED [22,23,24]. The states of hyperinsulinism and insulin resistance that are observed in women with PCOS may be responsible for their experience of increased levels of cravings. Furthermore, the desire to eat food rich in carbohydrates is intensified [25]. Women with PCOS who experience intensified food cravings are more vulnerable and more likely to demonstrate binge eating behavior. On the other hand, hyperinsulinism promotes the production of more androgens by the theca cells in the ovaries [26]. Moreover, hyperandrogenism observed in women with PCOS may promote both hyperinsulinism and insulin resistance [27]. It is also known that a high level of serum testosterone promotes increased food intake in males [28].

The hyperandrogenic state, which is one of the characteristic features of PCOS, is a common factor not only postnatally; it has also been commonly observed prenatally as one of the major factors contributing to the development of PCOS [29,30].

It is also possible that prenatal exposure to androgen excess may play a role in fetal brain organization and development. As a result, women who were prenatally exposed to elevated androgen hormone levels could be at risk of altered neuronal signaling and, as a consequence, undergo a distinct character and temperament development and have a higher frequency of personality and mental disorders. This seems to have been confirmed by some research studies. For example, prenatal exposure to androgen excess has been associated with increased impulsivity [31], reward sensitivity [32], elevated levels of aggression [33], and decreased social cognition [34]. All of these states are regarded as binge eating disorder risk factors [35]. In addition, craving carbohydrate-rich food, which thus promotes insulin hypersecretion, results in a fluctuation of serotonin levels which, in turn, influence a person’s well-being [36]. A lot of data indicate that BED may be successfully treated with both cognitive and behavioral and/or pharmacological treatment [37,38,39,40,41].

Considering the fact that women with PCOS are at a higher risk of binge eating behavior and its possible impact on results of the treatment, there is a need to screen women with PCOS for BED by gynecologists. The Questionnaire for Binge Eating Screening (QBES) may be a useful diagnostic tool. What is more, it does not require additional time during a medical visit as it is a self-rating questionnaire, in comparison to the DSM-5 questionnaire which has to be conducted by a physician and requires about 20 min.

## 5. Conclusions

Binge eating is a very frequent comorbid condition in Polish women with polycystic ovary syndrome; this constitutes a finding similar to findings made in other countries. BED may have an unfavorable impact on the course of PCOS; thus, it is necessary to screen women for the risk of developing BED and refer them for a psychiatric consultation and treatment once a preliminary diagnosis is made. The Questionnaire for Binge Eating Screening (QBES) has very high sensitivity and specificity, even when only the first two questions are asked; however, the replication of these results in a bigger sample would increase the value of this finding.

## Figures and Tables

**Table 1 ijerph-20-00546-t001:** The prevalence of BED symptoms of among subjects with and without BED in the study group.

DSM-5 Criteria for BED	PCOS with BED	PCOS without BED
A1. An episode of binge eating is characterized by eating, in a discrete period of time (for example, within any two-hour period), an amount of food that is definitely larger than most people would eat in a similar period of time under similar circumstances.	51 (100%)	17 (24%)
A2. An episode of binge eating is characterized by a sense of lack of control over eating during the episode (for example, a feeling that one cannot stop eating or control what or how much one is eating).	51 (100%)	10 (14%)
B1. Eating much more rapidly than normal.	29 (57%)	6 (8%)
B2. Eating until feeling uncomfortably full.	43 (84%)	12 (17%)
B3. Eating large amounts of food when not feeling physically hungry.	39 (76%)	8 (11%)
B4. Eating alone because of feeling embarrassed by how much one is eating.	21 (41%)	4 (6%)
B5. Feeling disgusted with oneself, depressed, or very guilty afterwards.	45 (88%)	9 (13%)
C. Marked distress regarding binge eating is present.	50 (98%)	10 (14%)
D. The binge eating occurs, on average, at least once a week for three months.	51 (100%)	6 (8%)
At least one symptom.	51 (100%)	25 (35%)
None of the symptoms.	0 (0%)	46 (65%)

**Table 2 ijerph-20-00546-t002:** Sensitivity and specificity of the Questionnaire for Binge Eating Screening (QBES).

	Sensitivity	Specificity
More than 0 positive criterion in the screening tool	72%	58%
More than 1 positive criterion in the screening tool	88%	67%
More than 2 positive criteria in the screening tool	100%	80%
More than 3 positive criteria in the screening tool	100%	91%
For both the 1st and 2nd question in the screening tool	98%	85%

## Data Availability

The data presented in this study are available on request from the corresponding author. The data are not publicly available due to current calculations to the next publication from this study.

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
