# Peer review of "Binge Eating Disorder—The Point Prevalence among Polish Women with Polycystic Ovary Syndrome and Validity of Screening Tool for This Population"

_ijerph, 2022, doi:10.3390/ijerph20010546_

Round 1
Reviewer 1 Report
This manuscript by Suchta et al. applies the DSM-5 criteria for binge eating disorder (BED) to patients with PCOS to determine whether there is an association between these conditions. The authors state that this is the first time this has been done for the Polish population. Additionally, the authors evaluated a shorter screening tool for BED, the Questionnaire for Binge Eating Screening (QBES) for its effectiveness in identifying patients with BED as diagnosed by the DSM-5 criteria. They find that the QBES is robust in this regard, and even only the first two questions show high sensitivity and specificity in identifying clinical BED. This study provides useful information that may make it easier for patients to be referred to proper treatment for BED, which may relieve some PCOS symptoms. However, there are a few suggestions and points of clarification required:
- I didn’t see any information about how BED was diagnosed from the DSM-5 criteria. Was there a threshold number of questions answered in the affirmative that are required for diagnosis? Could the authors please clarify how PCOS patients were initially categorized into the groups with or without BED?
- In the first column of Table 1, it would be helpful to label the questions according to the categories of the DSM-5 (A1, A2, B1, etc.) rather than (1, 2, 3,…).
- The methods say 122 patients took part in the study, but the numbers in Table 1 add up to 121. Please clarify.
- Though the QBES is only 4 questions, and the authors show that even asking the first two questions can be informative for indicating BED, the DSM-5 criteria are not that much more involved. Is there any information on how long it takes to go through the DSM-5 criteria vs. the QBES? Is the small amount of time saved by using the QBES meaningful in the clinic? Perhaps the authors can add their thoughts on this question to the Discussion.
Reviewer 2 Report
The introduction should be expanded by better describing the eating patterns frequently found in women with PCOS. In addition, a brief description of how these may be a cause or consequence of eating behaviour and possible eating disorders in women with PCOS may be useful (please see and quote 10.3390/ijms21218211, 10.1016/j.metop.2021.100123, 10.3390/nu14050955)
Lines 40-43: “Its diagnosis requires the presence of at least two out of three elements, i.e. biochemical and/or clinical hyperandrogenism, anovulatory menstrual cycles and/or rare ovulations, and a specific echogenicity of the ovaries in a pelvic ultrasound examination”
This definition of the Rotterdam diagnostic criteria seems a little too approximate (for example rare ovulations, specific echogenicity). Please define the three diagnostic criteria in a better way (in particular also in lines 71 -73, methods section)
Lines 65-67: “One-hundred twenty-two (122) out of 140 (87.1%) women with PCOS who had been 65 invited to participate took part in the study. Their age ranged between 16 and 45 (M = 26; 66 SD = 5.22)..”
These are already results, which is why I suggest moving them to the results section.
Lines 90-104: “Binge Eating Disorder was diagnosed according to the DSM-5 criteria..”
The BED criteria thus reported appear a bit confusing. I suggest the authors put them in a table to make the reading more fluent.
Lines 140-150: “The relatively low rate of refusals indicates that this group of PCOS women is relatively representative and large enough.”
I do not agree that these reasons justify the reported statement. Has the sample size been calculated?
Lines 153-155: “In our analysis, we report that binge eating behavior occurs about ten times more frequently among PCOS women when compared to non-PCOS women”.
Which analyses are the authors referring to? there is no data on the results and the quotation given does not seem to be from the same research group.
Lines: 156-157: “Our results are comparable to other publications, reporting that approximately 30% of PCOS women has BED [13,18]”
Also there, what results do the authors refer to?
Lines 171-173: “Animal studies have shown that prenatal exposure to androgen excess in female rat fetuses promotes the development of masculinized eating patterns in adulthood that include a reduced number of meals but increased meal size”
I don't think you can define an eating pattern as “masculinized”. Female rats exposed to high levels of androgens changed their eating habits, making them more similar to those observed in male rats. In any case, I do not believe that this study can support a possible role of androgens in the development of BED in women with PCOS.
The discussion lacks an argumentation regarding the questionnaire under study.
I think it is a bit of an exaggeration to define a questionnaire assessed on such a small population highly specific and sensitive. Please reduce the tone of the conclusions and focus more on the need for further well-designed studies to actually validate the use of this questionnaire.
Ensure consistency in the use of abbreviations.
Text revision by a native English speaker is highly necessary.
Reviewer 3 Report
Dear Authors,
Dear Authors,
The manuscript presents research conducted on the population of women in Poland regarding to estimate the prevalence of BED in PCOS patients and to assess the sensitivity and specificity of the Questionnaire for Binge Eating Screening (QBES) in PCOS patients.
The concept of the study is interesting and the results are promising.
Limitations:
(1) In my opinion, the research is limited by the study group, which should be more numerous for such boldly presented conclusions.
(2)The introduction should be extended with literature data in this scope.
How are the results presented in the work compared to data from other research centers in the world?
Author Response
Please see in the attachment

Round 2
Reviewer 2 Report
The authors have appropriately resolved all the issues raised. The only advice I leave to the authors is to use person-oriented language to deal respectfully with people living with chronic diseases. So instead of defining subjects as obese, it would be better to define them in one of the following ways: subjects with obesity of subjects living with obesity.
Reviewer 3 Report
Dear Authors,
Thank you for following my comments and suggestions, which significantly influenced the substantive value of the manuscript.